# Segmental Pulmonary Hypertension in Children with Congenital Heart Disease

**DOI:** 10.3390/medicina56100492

**Published:** 2020-09-24

**Authors:** Bibhuti B. Das, Benjamin Frank, Dunbar Ivy

**Affiliations:** 1Department of Pediatric Cardiology, Baylor College of Medicine, Texas Children’s Hospital Austin Specialty Care, Austin, TX 78759, USA; 2Department of Pediatric Cardiology, University of Colorado Health Science Center, Children’s Hospital of Colorado, Denver, CO 80045, USA; Benjamin.Frank@childrenscolorado.org (B.F.); Dunbar.Ivy@childrenscolorado.org (D.I.)

**Keywords:** pulmonary hypertension in children, congenital heart disease, segmental pulmonary hypertension, digital subtraction angiography

## Abstract

Segmental pulmonary hypertension is a complex condition in children that encompasses many congenital heart diseases including pulmonary atresia with ventricular septal defect, hemitruncus/truncus arteriosus with branch pulmonary artery stenosis, unilateral absent pulmonary artery, and several post-tricuspid shunt lesions. Multimodality imaging is required to confirm and assess pulmonary vascular disease in subjects with major aorto-pulmonary collaterals. We describe 3 children with complex congenital heart defects who have a variable degree of segmental pulmonary hypertension and discuss management strategies and the role of interventional and/or pulmonary hypertension targeted therapies.

## 1. Introduction

The segmental pulmonary hypertension (PH) is defined as pulmonary vascular disease (PVD) in one or more but not all segments of the lung. Each hypertensive segment may present with different degrees of PVD due to variable blood flow, local pressures, and different sources. Symptoms of segmental PH are often proportional to the severity of the ventilation-to-perfusion mismatch. A consensus statement to define and classify segmental PH is published in 2018 [1]. According to the statement, segmental PH is commonly encountered in patients with congenital heart diseases (CHD) such as tetralogy of Fallot (TOF) with pulmonary atresia and ventricular septal defect (VSD), hemitruncus arteriosus (one pulmonary artery is normally connected, but the other one arises from the aorta), truncus arteriosus with stenosis or hypoplasia of a single pulmonary artery or branches, unilateral absence pulmonary artery or isolated pulmonary artery of ductal origin, any large post-tricuspid shunts with peripheral pulmonary stenosis, and isolated peripheral pulmonary artery stenosis, for example, Alagille syndrome.

There are substantial differences in pathophysiology in various types of segmental PH compared with other types of PH. In segmental PH due to pulmonary atresia with VSD or truncus arteriosus, the right ventricle (RV) hypertrophy, dilation or dysfunction can result from large VSD and resultant systemic RV pressure. On the other hand, when RV is directly connected to the pulmonary circulation, the effect of segmental PH on RV is in addition to the load imposed by obstructive lesions such as branch pulmonary artery stenosis or RV-pulmonary artery conduit stenosis. It is essential to understand the anatomy of pulmonary arteries and major aorto-pulmonary collaterals (MAPCAs), as these influence the development of PH, response to therapy, long-term outcome and is key to decide when a patient can undergo bi-ventricular repair [1]. Although cardiac catheterization is the standard for assessing segmental PH, the calculation of pulmonary vascular resistance (PVR) is difficult and sometimes not helpful. It is essential to determine the hemodynamic of each segmental vessel. Multimodality imaging can be useful to guide invasive cardiac catheterization. In this paper, we describe 3 cases representing common causes of segmental PH in children with CHD and the challenges in their management.

Case-1: The patients is a 17-year-old female who was diagnosed at the age of one month with TOF and pulmonary atresia with discontinuous and hypoplastic pulmonary arteries. She initially underwent a Mee shunt (a central shunt between the ascending aorta and the main pulmonary artery) at the age of 2 months followed by 4 mm right modified Blalock-Taussig (BT) shunt at 5 months. She then had a left modified BT shunt and left unifocalization at 10 months. She had first cardiac catheterization at 18 months due to hypoxia and the left modified BT shunt was found completely occluded and could not be recanalized. Her next cardiac catheterization was at 5 years and showed MAPCAs with patent right modified BT shunt and proximal stenosis of the right middle lobe and right lower lobe pulmonary arteries. There were multiple MAPCAs from the descending aorta to the right upper and lower lobes and left lung with proximal stenosis. She had another catheterization at 7 years and angioplasty of the MAPCAs was performed in the right lower lobe and left lower lobe. She then lost to follow-up and admitted to the local hospital at age 15 years with brain abscess. Her brain abscess was treated and resolved.

At age 17 years, she presented to the cardiology clinic for worsening cyanosis (SpO2 in 60%), erythrocytosis (hematocrit 72%), easy fatigability, and decreased exercise tolerance (NYHA Class III). Her weight 47.9 Kg, height 150 cm, and body mass index (BMI) was 21.59 kg/m². She had clubbing of fingers and toes, but no edema. Her lungs were clear to auscultation. She had normal first heart sound, single second heart sound but loud, with a faint continuous murmur on right back. There was no hepatosplenomegaly. Her electrocardiogram (ECG) showed sinus rhythm, right axis deviation, and right ventricular hypertrophy (Figure 1). Chest X-Ray showed bilateral heterogeneous vascular markings (Figure 2). Echocardiogram showed a large perimembranous VSD with aortic override and right-to-left shunt across VSD. Bi- ventricular systolic function was normal. There was mild aortic valve regurgitation. Her laboratory test was significant for a normal brain-type natriuretic peptide level of 65 pg/mL (normal, <100 pg/mL).

She underwent computed tomography (CT) angiogram of the chest which showed occluded right BT shunt and MAPCAs with a variable degree of stenosis supplying to bilateral lung fields (Figure 3). There were hypoplastic branch pulmonary arteries which were of ductal origin. She underwent cardiac catheterization and balloon angioplasty and stenting of MAPCAs. Her SpO2 increased from 60 to 75%. Selective angiogram of MAPCAs on the right side with digital subtraction showed the abnormal distal arborization of pulmonary vasculature suggesting vascular remodeling and PVD in the right upper lobe (Figure 4A) compared to the lower lobe (Figure 4B). She was discharged on aspirin, iron supplementation, and oxygen. On the latest follow-up 3 months after the interventional procedure, her functional capacity improved from NYHA class III to Class II.

Discussion (case-1): Tetralogy of Fallot, pulmonary atresia with VSD, and MAPCAs is a rare congenital cardiac anomaly and one of the most complex defects to manage surgically. There are 3 types based on pulmonary blood supply: Group I had well-formed native pulmonary arteries with MAPCAs, group II had hypoplastic pulmonary arteries with MAPCAs, and group III had only MAPCAs without native pulmonary arteries [2]. The three groups were further subcategorized into those patients with protected MAPCAs with proximal stenosis, and those with unprotected MAPCAs (unobstructed flow). The natural history of the MAPCAs follows a course of progressive stenosis and occlusion. Hypoxemia or inadequate blood flow are described to trigger postnatal maladaptation or underdevelopment of the distal pulmonary vessels in vitro studies as well as in people native to high altitudes [3,4,5]. There is concern that even if flow-induced growth of the proximal pulmonary arteries can be achieved, the development of the distal pulmonary vessels remains unclear [6,7]. Thus, the sooner these collateral arteries are unifocalized and normal physiologic circulation is established, the greater is the number of healthier lung segments that can be incorporated into the pulmonary circulation and better is the long-term outcomes [8]. An area of the lung may receive blood supply from the native pulmonary arteries or MAPCAs or in combination. Depending on these variations, these patients are initially seen with either cyanosis, caused by insufficient pulmonary blood flow, or congestive heart failure or PH, caused by excessive pulmonary flow. Our patient has a type II pulmonary blood supply and initially underwent a central shunt (Mee shunt) followed by BT shunts to promote the growth of hypoplastic pulmonary arteries. However, due to lack of regular follow-up and compliance, there were multiple issues with BT shunt occlusions and failure of growth of pulmonary arteries and worsening stenosis of MAPCAs are prohibitive to achieve the ultimate plan to perform the bi-ventricular repair.

Evaluation of segmental PH in subjects with MAPCAs is best done using a multimodality approach. Three-dimensional (3D) imaging technology with helical computed tomography (CT) and electron beam CT and cardiac magnetic resonance imaging (CMR), has progressed significantly, allowing 3D visualization of pulmonary arteries and MAPCAs which is crucial for preoperative planning [9]. The calculation of PVR is impossible as the MAPCAs originating from aorta have the same systemic saturation and was difficult to quantify the pulmonary blood flow. However, the measurement of blood flow in MAPCAs is possible in the operating room using a modified pump circuit [10]. Noninvasively, phase contrast cine CMR blood flow can be measured accurately in branch pulmonary arteries and MAPCAs in single ventricle physiology [11]. Phase contrast CMR can be useful for screening patients with unilateral stenosis and contralateral increases in PVR by estimating the differential regurgitant fraction of the branch pulmonary artery that cannot be identified with net blood flows alone [12]. Phase contrast CMR analyzes blood flow data at regions of interest throughout all phases of the cardiac cycle. Forward, regurgitant, and net flows were then automatically calculated from the resulting time-volume curves. The percentage regurgitant flow through a region of interest is defined as follows: (reverse flow/forward flow) × 100. Fractional branch pulmonary artery blood flow is calculated from: (net blood flow in branch pulmonary artery/net total pulmonary blood flow) × 100. Differential regurgitant fraction a function of the relative PVR and the presence of branch pulmonary artery stenosis or size discrepancy. During cardiac catheterization, selective angiogram of MAPCAs can demonstrate the status of distal pulmonary vessel arborization and perfusion defects using digital subtraction technique [13].

Pediatric patients and adults with TOF, PA, and MAPCAs who received bosentan and sildenafil have improved symptomatically [14,15,16,17,18]. However, there is a need for further study to determine whether PH medications provide a long-term survival benefit for these patients.

Case-2: A 7-year-old girl with a history of truncus arteriosus and severe truncal valve stenosis, s/p Rastelli procedure for truncus arteriosus at 4 months. She developed mild to moderate truncal valve regurgitation but truncal valve stenosis became progressively severe. She also developed mild to the moderate right ventricle to pulmonary artery conduit stenosis and increasing conduit regurgitation. At 9 months, she was admitted with tachypnea, lethargy, and decreased oral intake. She then underwent aortic valve and root replacement, right ventricular outflow tract reconstruction, a transannular patch with a pericardial mono cusp valve, and pulmonary artery reconstruction.

At age 7, her physical examination revealed weight 21.5 kg, height 114 cm, BMI was 16.4 kg/m^2^. She had clear lungs. Her cardiac examination was remarkable for normal intensity single first sound, widely split second sound, grade 4/6 systolic ejection murmur at mid-left sternal border radiating along clavicles, and radiate to back, and up the left and right sides of her neck. She also had a grade 3/6 decrescendo medium pitched diastolic murmur (Austin Flint) at mid-left sternal border radiating to the apex and up the neck. There was no hepatosplenomegaly.

Her ECG showed sinus rhythm and left ventricular hypertrophy (Figure 5). Chest X-Ray showed cardiomegaly with increased bronchovascular markings (Figure 6). Echocardiogram showed mildly dilated and hypertrophied RV with a diastolic septal flattening, normal-sized normal left ventricle, and normal biventricular systolic function. There was severe pulmonary valve regurgitation with mild stenosis of the RV to pulmonary artery conduit with a mean gradient of 40 mmHg. Her CT angiogram (Figure 7) showed no residual ventricular septal defect and mildly dilated aortic root 18 × 17 × 14 mm), moderately to severely dilated ascending aorta (29 × 27 mm) and arch of aorta (27 × 26 mm). Right (11 × 9 mm) and left (15 × 11 mm) pulmonary arteries were mildly dilated. There are stenosis of the right pulmonary artery in distal branches. There were no large MAPCAs. She remained on aspirin only. Cardiac catheterization showed dilated branch pulmonary arteries but stenosis of right branch pulmonary arteries (Figure 8) and hemodynamics revealed normal total PVR. The lung perfusion study showed 65% of flow to left lung compared to 35% to right lung. As there was increased blood flow to the left lung compared to the right lung, changes in the distal pulmonary vasculature were significant in the left lung as shown in Figure 9A compared to right pulmonary vasculature (Figure 9B).

Discussion (case-2): Truncus arteriosus is characterized by a common arterial trunk, giving rise to both the systemic and pulmonary circulation in various patterns. Truncus arteriosus is frequently associated with other cardiac and great vessel anomalies which are present in 34.8% of cases, such as right aortic arch (25–30% of cases), interrupted aortic arch, aberrant right subclavian artery, abnormal coronary arteries, atrial septal defect, tricuspid atresia, and double aortic arch [19]. Two classifications have been proposed for truncus arteriosus. Collett and Edwards classified truncus arteriosus into 4 types [20]. Van Praagh classified truncus arteriosus into 2 types based on the presence (type A) or absence (type B) of a VSD with the latter type being rare. Type A was reclassified into 4 types [21]. In type A1, the main pulmonary artery is arising from a common trunk and aortopulmonary septum is partially formed. The aortopulmonary septum is completely absent in type A2 and right and left pulmonary arteries directly forming from truncus. In Type A3 one pulmonary artery is absent and that lung is supplied by collateral vessels or a pulmonary artery from a patent ductus arteriosus or MAPCAs from the descending aorta. Type A4 is defined not by the pattern of origin of the pulmonary arteries but by the coexistence of hypoplasia, coarctation, atresia, or absence of the aortic arch. The absence of pulmonary artery stenosis leads to the early development of bilateral PH.

In this case, mild stenosis of distal right pulmonary artery branches lead to relatively less blood supply on the right side compared to increased pulmonary blood flow to the left lung, leading to a flow-related change in pulmonary vasculature (Figure 9B). The perfusion scan confirmed 65% of flow into left and 35% to the right. The patient is scheduled for genetic evaluation of aortopathy as the cause of aortic root and arch is unknown. No targeted PH therapy is started as this could cause increased flow to the unobstructed pulmonary segments and further increase flow-related PVD. She is scheduled for a repeat CT angiogram in near future for possible intervention of left branch pulmonary artery stenosis.

Case-3: A 12-year-old immigrant from Nepal, who had had a heart murmur since childhood, presented after recent-onset dyspnea on exertion and several blue spells. A physical examination revealed a resting oxygen saturation of 98% on room air, blood pressure of 104/70 mmHg, and a regular heart rate of 76 beats/min. There was a normal first heart sound, a loud second sound, and a grade 4/6 systolic ejection murmur with a palpable thrill along the left mid-sternal border. There was no evidence of peripheral or central cyanosis, but clubbing was present. The patient’s hemoglobin concentration was 18 g/dL and hematocrit was 68%. An electrocardiogram showed sinus rhythm, right axis deviation, right atrial enlargement, and biventricular hypertrophy (Figure 10). He underwent a cardiac catheterization and showed the right pulmonary artery arising from the aorta and no left pulmonary artery was seen from aortogram (Figure 11) and the left pulmonary artery was arising from the pulmonary trunk and connected to the RV (hemitruncus). There was loss of normal pulmonary vascular arborization and pruning of distal pulmonary vasculature suggestive of PVD in the right lung because of excessive pulmonary blood flow and systemic pressure as the right pulmonary artery was arising from the aorta.

Discussion (case-3): The anomalous origin of the right pulmonary artery, is thought to result from failure of development of the right sixth arch and persistence of the right fifth arch. Infants who are diagnosed with this anomaly early and undergo early surgical repair, have relatively better outcomes. Children who do not undergo surgical correction have a 70% first-year mortality rate, and 30% of the infants die within 3 months. Early repair is carried out to avoid the PVD of the lung [22]. If left untreated, as one pulmonary artery is derived from the aorta, segmental PH can develop due to pressure and volume overload. Digital subtraction angiography can be useful to document perfusion defects of the peripheral lung and diagnosis of PVD as previously shown in children with PH due to bronchopulmonary dysplasia [13]. No targeted PH therapy is used in this case as this will worsen the PVD on the right side and only supportive care including aspirin, and iron supplementation is offered.

## 2. Conclusions

Segmental PH is a complex condition that encompasses a wide range of CHD and its management requires tertiary expertise both in CHD and PH. There have been early attempts to modify PVD by targeted therapies after optimal pulmonary rehabilitation by catheterization and surgery. Some experts have concerns that pulmonary vasodilation could further volume load the heart by significantly augmenting pulmonary blood flow in bronchopulmonary segments where there is no stenosis of MAPCAs or pulmonary arteries and can result is reperfusion lung injury and secondary PH. There have been few publications reporting on the beneficial effects of bosentan and sildenafil in children and adults with segmental PH [14,15,16,17,18]. Certainly, there is the need for reproduction of these results and currently, there is no recommendation on the use of these agents in all cases of segmental PH. However, given the broad heterogeneity and relatively small number of patients and the difficulty to calculate PVR as a strong hemodynamic endpoint, randomized and adequately powered trials are unlikely to be realistic. Given the severity of the disease and the functional impairments often seen in the presence of segmental PH, we believe it is reasonable to consider targeted PH therapy in certain cases. Digital subtraction angiography may be useful as a diagnostic tool to delineate the distal pulmonary vasculature and PVD.

## Figures and Tables

**Figure 1 medicina-56-00492-f001:**
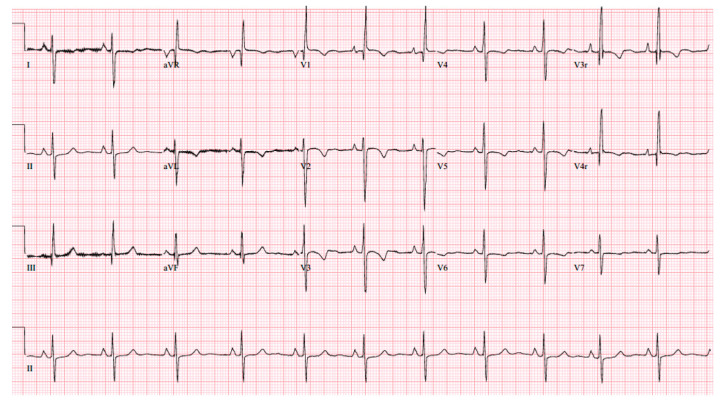
12-lead electrocardiogram (case-1). Sinus rhythm with right atrial enlargement and right ventricle hypertrophy.

**Figure 2 medicina-56-00492-f002:**
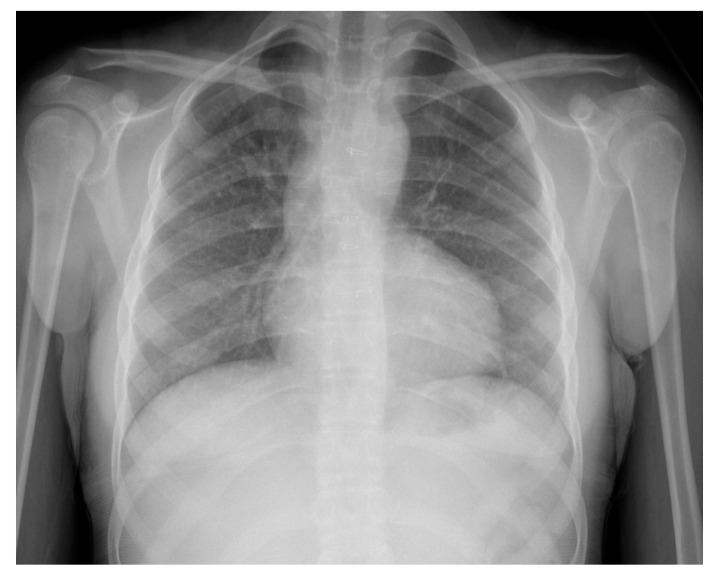
Chest X-ray (Case-1). “Boot shaped” heart and variable perfusion of the lung parenchyma.

**Figure 3 medicina-56-00492-f003:**
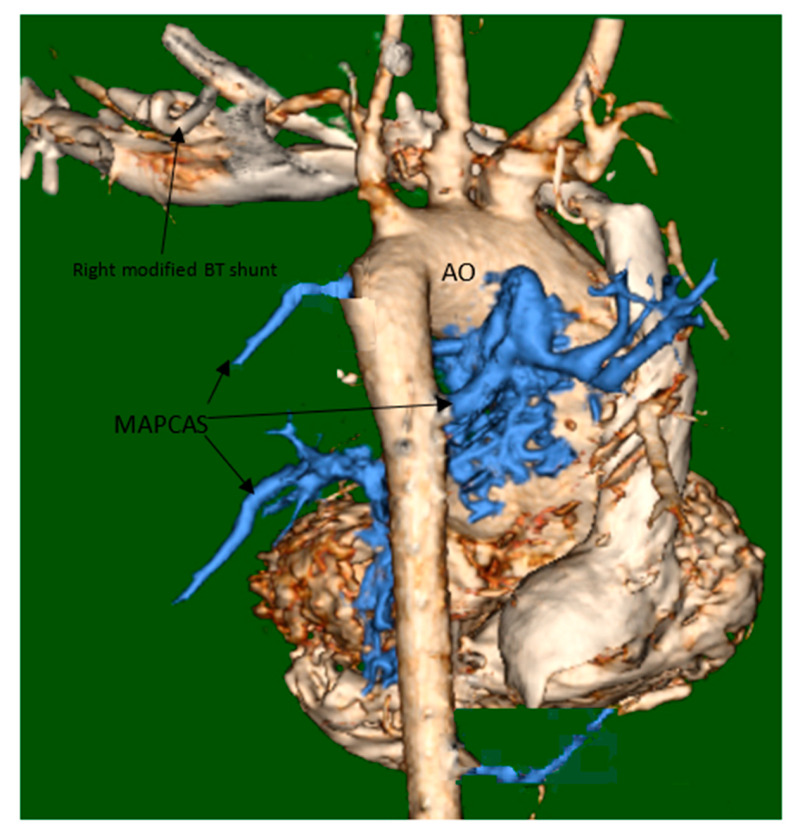
CT angiogram and 3-D reconstruction. 3-D visualization major aorto-pulmonary collaterals (MAPCAs). CT: computed tomography. AO: aorta. BT: Blalock-Taussig.

**Figure 4 medicina-56-00492-f004:**
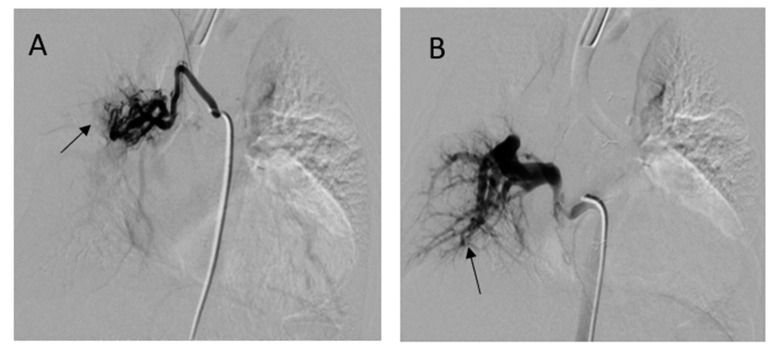
Digital subtraction angiography of selective MAPACs. (**A**) MAPCA supplying the right upper lobe showed absence of normal arborization (arrow) and (**B**) the lower MAPCA showed relatively normal distal pulmonary vascular arborization pattern in right lower lobe (arrow).

**Figure 5 medicina-56-00492-f005:**
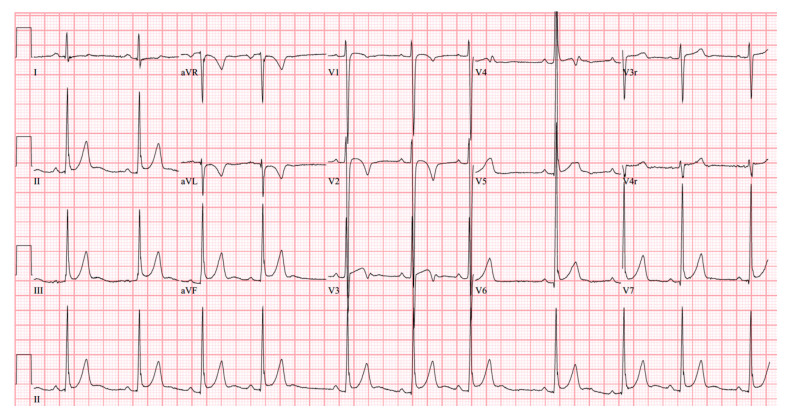
12-lead electrocardiogram (Case-2). Sinus rhythm with left ventricular hypertrophy and early repolarization pattern.

**Figure 6 medicina-56-00492-f006:**
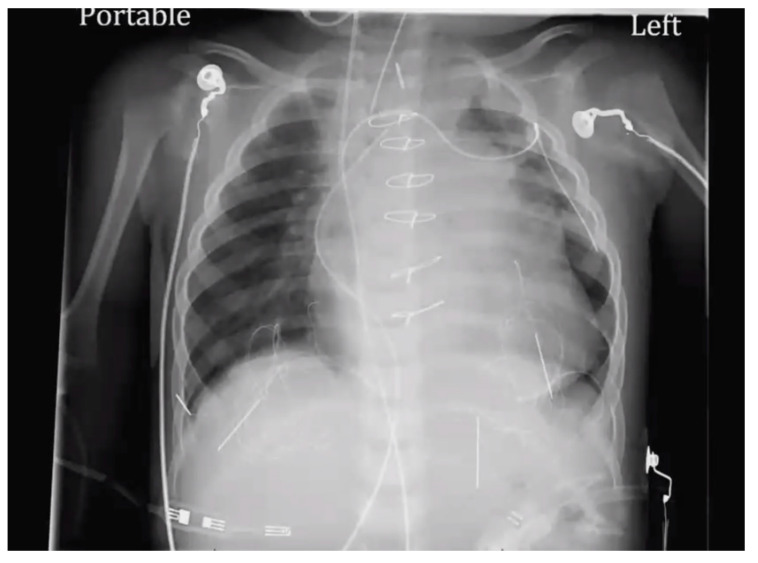
Chest X-ray (Case-2). Cardiomegaly, left lower lobe atelectasis.

**Figure 7 medicina-56-00492-f007:**
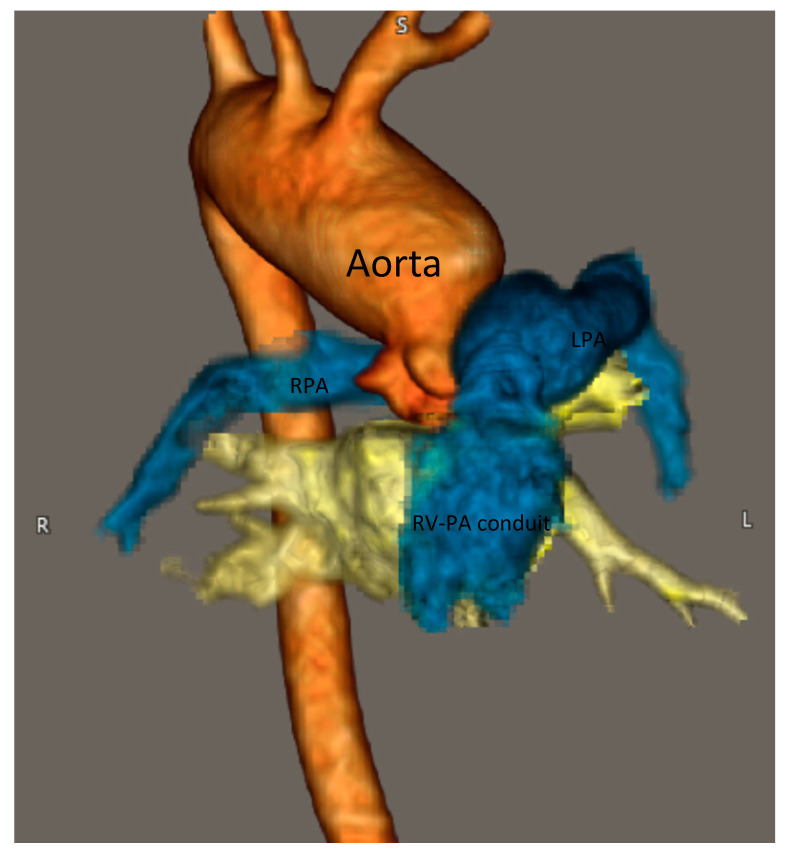
3-D CT reconstruction of aorta and pulmonary artery. Right aortic arch with mirror image branching, dilated aortic root and aortic arch, s/p Rastelli procedure (right ventricle-pulmonary artery (RV-PA) conduit) for truncus repair, right pulmonary artery (RPA) and left pulmonary arteries (LPA) mildly dilated diffusely, focal distal RPA stenosis. No large aortopulmonary collaterals.

**Figure 8 medicina-56-00492-f008:**
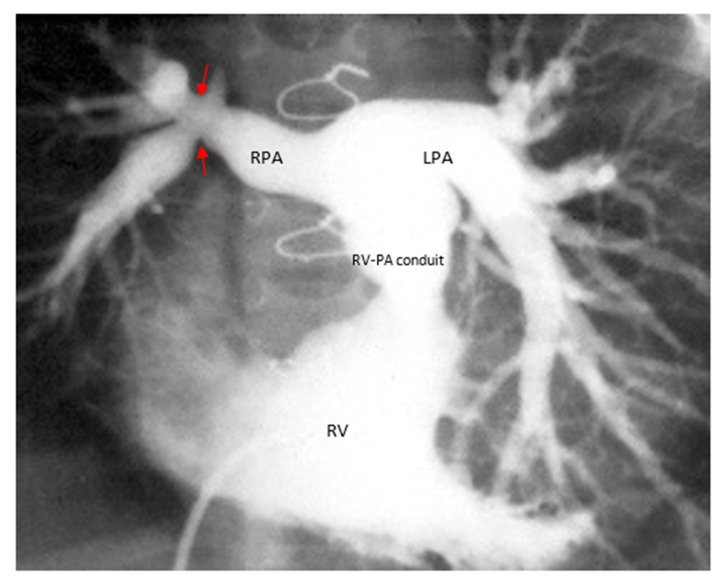
Right ventricle (RV) angiogram. Peripheral stenosis (arrows) of the right branch pulmonary arteries (RPA) and relatively large blood flow to the left lung through a large dilated left pulmonary artery (LPA).

**Figure 9 medicina-56-00492-f009:**
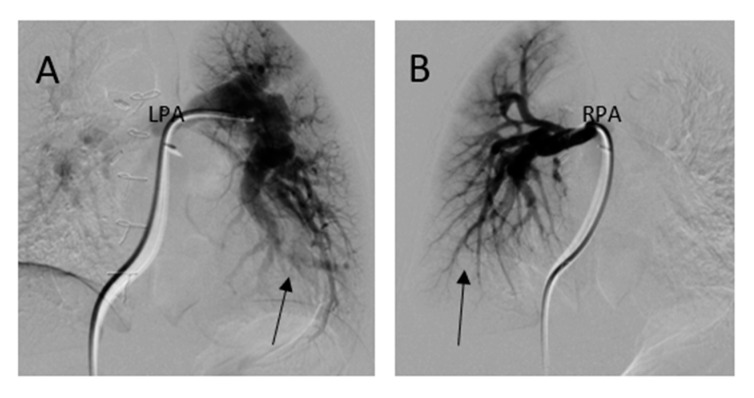
Digital subtraction angiography of left pulmonary artery (LPA) (**A**) compared right pulmonary artery (RPA) (**B**); (**A**) Arrow pointing to pruning of distal pulmonary vasculature on left; (**B**) Arrow pointing to normal distal pulmonary vascular arborization pattern on right.

**Figure 10 medicina-56-00492-f010:**
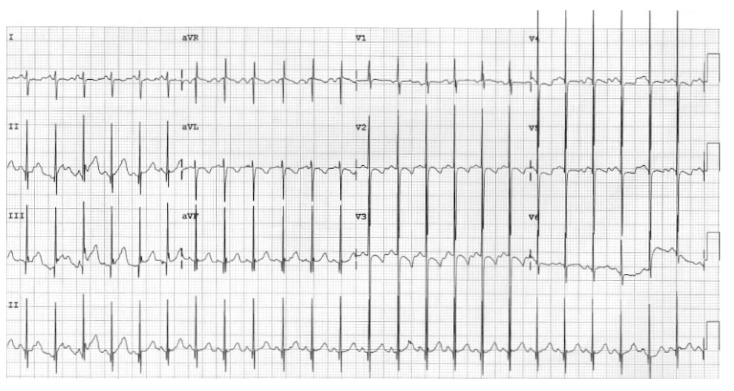
12-lead electrocardiogram (Case-3). Sinus rhythm and Katz-Wachtel phenomenon suggesting bi-ventricular hypertrophy.

**Figure 11 medicina-56-00492-f011:**
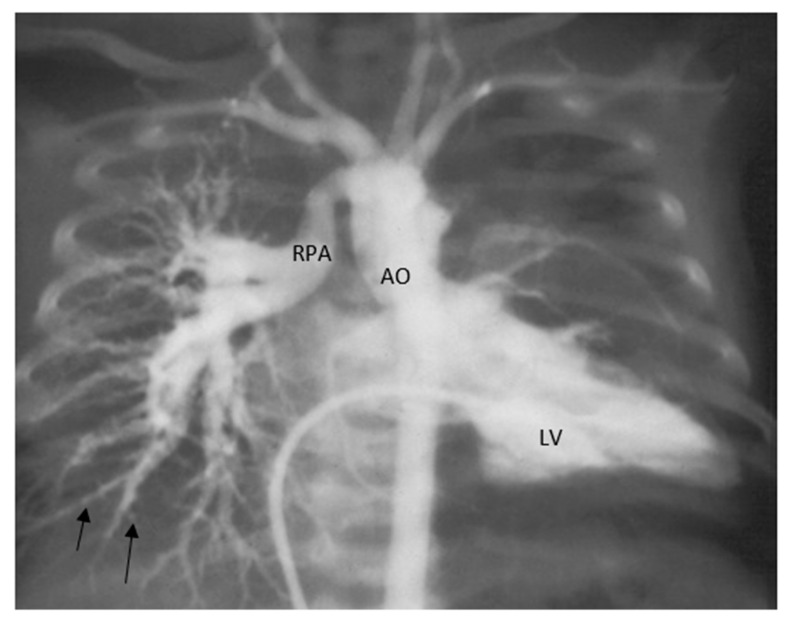
Left ventricle (LV) angiography showing right pulmonary artery (RPA) arising from aorta (AO). Arrows pointing to pruning of distal vasculature on right lung.

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
