# Peer review of "Segmental Pulmonary Hypertension in Children with Congenital Heart Disease"

_medicina, 2020, doi:10.3390/medicina56100492_

Round 1
Reviewer 1 Report
In this brief report, the authors present three cases of patients with congenital heart defects (CHD) and segmental pulmonary hypertension (PH). The manuscript has several English language syntax errors that require careful revision.
The introduction lists the most common CHDs that may be associated with segmental PH, but does not provide any context on why segmental PH is difficult to diagnose and important to treat.
The authors do not report any unusual case presentations and do not offer any novel or groundbreaking management modalities.
The figure captions should include a brief description of the patients' diagnosis and the images need to identify the anatomic structures.
Author Response
In this brief report, the authors present three cases of patients with congenital heart defects (CHD) and segmental pulmonary hypertension (PH). The manuscript has several English language syntax errors that require careful revision.
- Thank you. The manuscript is revised with correction of grammar.
The introduction lists the most common CHDs that may be associated with segmental PH, but does not provide any context on why segmental PH is difficult to diagnose and important to treat.
- We added a second paragraph in the introduction describing the pathophysiology and difficulties to diagnose segmental PH.
The authors do not report any unusual case presentations and do not offer any novel or groundbreaking management modalities.
- We have shown that multi-modality imaging is helpful to delineate the collaterals and helpful before invasive cardiac catheterization. The novel finding is demonstration of distal pulmonary vasculature by digital subtraction angiography of selective MPCAs as in case-1 (Figure 4A and 4B) and selective pulmonary artery in Case-2 (Figure-9A and B).
The figure captions should include a brief description of the patients' diagnosis and the images need to identify the anatomic structures.
- Thank you. Images have been re-labeled and captions of each figure are revised.
Reviewer 2 Report
These an interesting case series, elegantly demonstrating the presentation and management of segmental PH. The description of cases is very good, with minor English issues.
Comments:
- Line 30: Delete “The”: Segmental ….
- It may be worthwhile mentioning that the Mee shunt is also called a Waterstone shunt, or at least describe it briefly (central shunt, ascending aorta to PA) for physicians who do not use this term
- Some experts tend to avoid the term polycythaemia in congenital heart disease to avoid confusion with polycythaemia rubra vera. The term erythrocytosis may be preferable.
- Line 69: Biventricular?
- Line 71: Shunt, not shunts
- Line 72: MAPCAs
- Line 76 showed abnormal
- Fig 1: Why were right precordial and V7 tracings taken?
- Fig 3 would benefit from arrows pointing to MAPCAs and other structures.
- Fig 4 legend: MAPCAs supplying the right lung
- Line 125: Do you not expect cyanosis in all? Obligatory mixing in the heart? Not just because of reduced Qp.
- Line 130: the patient was lost to fu at the age of 7y. When were you planning unifocalization for?
- Line 145: please explain the concept of regurgitant fraction of a branch pulmonary artery
- Line 155, please state the type of valve used.
- Line 198: Large flow? Increased?
- Line 177: Please state the PA pressures in both lungs, and describe the RPA stenosis shown in Fig 8. Why was the RPA stenosis not seen in CT? Why did Echo not provide TR gradient? Please focus on the case in the discussion (less so on the classification of Truncus) describing the findings and how these suggest segmental PH. What treatment was started? Would you repair the RPA stenosis, and how is this decision influenced by haemodynamics and particularly by the pressure in the distal RPA and LPA? Why was the patient so unwell overall?
- Case 3: Was there an LPA originating from the RV or elsewhere? Was there a main PA seen on echo? Why was there an ejection systolic murmur? The description of the case is incomplete and there is nothing on management, why it is too late to repair etc.
- The absence of RPA stenosis with this anatomy and age suggests that there is established pulmonary vascular disease and an echo would probably suffice. Digital subtraction is only useful to identify perfusion defects, due to collateral circulation? Developmental? Other cause? PE very unlikely and contributes little to the diagnosis of PVD.
- The conclusions are appropriate.
Author Response
These an interesting case series, elegantly demonstrating the presentation and management of segmental PH. The description of cases is very good, with minor English issues.
- Thank you so much.
Comments:
- Line 30: Delete “The”: Segmental …. Done
- It may be worthwhile mentioning that the Mee shunt is also called a Waterstone shunt, or at least describe it briefly (central shunt, ascending aorta to PA) for physicians who do not use this term - Added after Mee shunt (A central shunt is an anastomosis between the ascending aorta and the main pulmonary artery)
- Some experts tend to avoid the term polycythaemia in congenital heart disease to avoid confusion with polycythaemia rubra vera. The term erythrocytosis may be preferable. - Thank you. Changed to erythrocytosis.
- Line 69: Biventricular? - Both ventricular systolic function is changed to Bi-ventricular systolic function
- Line 71: Shunt, not shunts - Thank you for the pick-up. Corrected the typo.
- Fig 1: Why were right precordial and V7 tracings taken? - This is a standard electrocardiogram in all patients in our center.
- Fig 3 would benefit from arrows pointing to MAPCAs and other structures. - Thank you, Arrows added to Figure-3 and other figures.
- Fig 4 legend: MAPCAs supplying the right lung - Figure-4 legend is revised.
- Line 125: Do you not expect cyanosis in all? Obligatory mixing in the heart? Not just because of reduced Qp. - Yes, cyanosis is due to obligatory mixing in the ventricles due to VSD
- Line 130: the patient was lost to Fu at the age of 7y. When were you planning unifocalization for? - The patient has irregular follow-up due to compliance issues. Unifocalization should have been done early in toddler if there was good rehabilitation of both PAs.
- Line 145: please explain the concept of regurgitant fraction of a branch pulmonary artery - Added a line: Differential regurgitant fraction a function of the relative PVR and the presence of branch pulmonary artery stenosis or size discrepancy.
- Line 155, please state the type of valve used. - Added: pericardial mono-cusp valve
- Line 198: Large flow? Increased? - Increased flow, corrected
- Line 177: Please state the PA pressures in both lungs, and describe the RPA stenosis shown in Fig 8. Why was the RPA stenosis not seen in CT? Why did Echo not provide TR gradient? Please focus on the case in the discussion (less so on the classification of Truncus) describing the findings and how these suggest segmental PH. What treatment was started? Would you repair the RPA stenosis, and how is this decision influenced by hemodynamics and particularly by the pressure in the distal RPA and LPA? Why was the patient so unwell overall? - Thank you. This portion is revised. CT did show focal distal RPA branches. The patient had multiple surgeries. First Rastelli operation is for truncus repair. Then Aortic root replacement. She had other issues such as compression of right bronchus with dilated aortic arch for which aortopexy is done. For all these reasons, she was sick. No intervention on right PA is done yet, but plan to follow with CTA in near future.
- Case 3: Was there an LPA originating from the RV or elsewhere? Was there a main PA seen on echo? Why was there an ejection systolic murmur? The description of the case is incomplete and there is nothing on management, why it is too late to repair etc. - Thank you. This has been corrected with addition of this line: the left pulmonary artery was arising from the pulmonary trunk and connected to the RV (hemitruncus) (added as 377 in revised manuscript).
- The absence of RPA stenosis with this anatomy and age suggests that there is established pulmonary vascular disease and an echo would probably suffice. Digital subtraction is only useful to identify perfusion defects, due to collateral circulation? Developmental? Other cause? PE very unlikely and contributes little to the diagnosis of PVD. - Thank you. DSPA is removed and the description has been revised.
- The conclusions are appropriate. - Thank you.
Round 2
Reviewer 1 Report
The authors have extensively revised the manuscript which has improved the clarity and readability.
The authors have addressed this reviewer's comments.
In the last sentence of case 1, the authors write "...her functional capacity improved from NYHA class II to Class II" Did they intend to say from class III to class II?
Author Response
Response to reviewer:
In the last sentence of case 1, the authors write "...her functional capacity improved from NYHA class II to Class II" Did they intend to say from class III to class II?
- Thank you for the pick-up, corrected.
Reviewer 2 Report
The paper has improved significantly
Minor comments:
- Fig 4a, I am not sure the black arrow points to a BP shunt, should it not be connecting to the blue PAs?
- Line 119, please give normal value for BNP.
- Line 252m, phrase missing verb, still not sure what differential regurgitant fraction is. Why should blood regurgitate from PAs? Do you mean relative distribution of pulm blood flow?
- Line 288 stenoses
- Line 292: please give formula for total PVR in the presence or RPA stenosis
- Line 518, I disagree that PH therapy will worse PVD, is there any evidence to support this strong statement?
- Line 550: impairment
- Reference 1 is incomplete
Author Response
The paper has improved significantly
- Thank you.
Minor comments:
- Fig 4a, I am not sure the black arrow points to a BP shunt, should it not be connecting to the blue PAs? - The black top arrow points to the right modified BT shunt and hypoplastic right PA, which has not grown. The blue vessels are MAPCAs.
- Line 119, please give normal value for BNP. - Thank you. We added the Normal BNP value which is <100 pg/L in parenthesis.
- Line 252m, phrase missing verb, still not sure what differential regurgitant fraction is. Why should blood regurgitate from PAs? Do you mean relative distribution of pulm blood flow? - Thank you. Phase contrast CMR data analyzes contouring regions of interest throughout all phases of the cardiac cycle. Forward, regurgitant, and net flows were then automatically calculated from the resulting time-volume curves. The percentage regurgitant flow through a region of interest is defined as follows: (reverse flow/forward flow) × 100. Fractional branch pulmonary artery blood flow is calculated from: (net blood flow in branch pulmonary artery/net total pulmonary blood flow) × 100.
- Line 288 stenoses - Thank you for the pick-up. Corrected.
- Line 292: please give formula for total PVR in the presence of RPA stenosis- - - This has been challenging to calculate total PVR in presence of multiple stenoses. CMR is useful and the formula for calculation of regurgitant flow as described in the previous query.
- Line 518, I disagree that PH therapy will worse PVD, is there any evidence to support this strong statement? - The effects of PH therapy in segmental PH remains a matter of debate, while a number of small studies showed symptomatic improvement (references cited 17-18), there are a number of cases where therapies were not tolerated (References cited: 15-16).
- Line 550: impairment - Corrected
- Reference 1 is incomplete - Reference #1, typos corrected.